# Piezoelectric Charge Coefficient of Halide Perovskites

**DOI:** 10.3390/ma17133083

**Published:** 2024-06-23

**Authors:** Raja Sekhar Muddam, Joseph Sinclair, Lethy Krishnan Jagadamma

**Affiliations:** Energy Harvesting Research Group, School of Physics & Astronomy, Scottish Universities Physics Alliance (SUPA), University of St Andrews, North Haugh, St Andrews KY16 9SS, UK; rsm22@st-andrews.ac.uk (R.S.M.); js450@st-andrews.ac.uk (J.S.)

**Keywords:** halide perovskite thin films, quasi-static measurements, d_33_ coefficient, halide perovskite crystals, 2D-perovskites

## Abstract

Halide perovskites are an emerging family of piezoelectric and ferroelectric materials. These materials can exist in bulk, single-crystal, and thin-film forms. In this article, we review the piezoelectric charge coefficient (d*_ij_*) of single crystals, thin films, and dimension-tuned halide perovskites based on different measurement methods. Our study finds that the (d*_ij_*) coefficient of the bulk and single-crystal samples is mainly measured using the quasi-static (Berlincourt) method, though the piezoforce microscopy (PFM) method is also heavily used. In the case of thin-film samples, the (d*_ij_*) coefficient is dominantly measured by the PFM technique. The reported values of d*_ij_* coefficients of halide perovskites are comparable and even better in some cases compared to existing materials such as PZT and PVDF. Finally, we discuss the promising emergence of quasi-static methods for thin-film samples as well.

## 1. Introduction 

Ambient energy harvesting has always garnered research attention as an alternative and sustainable powering method to meet our ever-increasing energy demand [1,2,3]. The anticipated future technological advancements such as the Internet of Things, wearables, and the international efforts to tackle the climate crisis (Net-Zero 2050) have led to unprecedented global research attention to the various ambient energy harvesting methods. Ambient energy harvesting refers to the energy conversion processes in which different forms of energy available in our surroundings (mostly wasted forms) are converted to a usable form of electricity [4,5]. Depending on the scale of harvestable energy, these electricity generation processes are classified as (a) microscale and (b) macroscale energy harvesting. In macroscale energy harvesting, large-scale sources such as sunlight, water flow, ocean waves, heat from the sun, wind, and the motion of large automotive vehicles such as trains, trucks, etc., are considered. In the case of microscale energy harvesting, the harvestable energy sources are relatively low-power, such as ambient light inside buildings, electromagnetic waves such as RF, vibrations from household appliances and industrial equipment, human walking, human movements, air-flow movements in HVAC systems etc. [6,7]. Ambient energy harvesting uses different physical phenomena such as the photovoltaic effect, electromagnetic induction, thermo-electric effect, pyroelectricity, piezoelectricity, triboelectricity, and magnetostriction in generating useful forms of electricity. Among these, the present review is particularly focussed on mechanical energy harvesting using the piezoelectric method with a particular emphasis on its application in the Internet of Things (IoT) [3].

The IoT is a smart network of internet-connected electronic and electrical devices that can communicate with each other and respond rapidly in real time. It is considered as the technology linking and integrating the digital and physical world. In the next 10–15 years, the number of connected devices in the IoT is predicted to range from 75 billion to more than 1 trillion [8]. Wireless sensors are the most fundamental components in the IoT, and they should have the ability to operate for long periods for continuous process monitoring and data transfer. Currently, they are mainly powered by batteries, and this reliance on batteries causes service interruptions, limits the functionality and deployability of IoT systems, and is not a practical powering method when these technologies have grown to their full potential (200 million battery replacements per day), and poses an environmental challenge due to the disposal of dead batteries as well [9]. Hence, the development of alternate powering methods other than batteries and wired power supply from the electrical grid is necessary for these wireless sensor platforms in the IoT. The self-powering of these wireless sensor platforms using ambient energy harvesters bestows them the ability to be functional for long periods autonomously [10,11]. 

Among the various ambient energy harvesting technologies available, photovoltaics and piezoelectricity are particularly significant because of the high density of harvestable ambient energy available (100 mW/cm^2^ for sunlight and 800 µW/cm^3^ for piezoelectricity) [12] and the high efficiency of the conversion to electrical power. State-of-the-art photovoltaic devices are capable of converting light to electricity with a 25% conversion efficiency, and within the last 30 years, several new photovoltaic materials other than the debut silicon solar cells have also emerged, such as CdTe, CIGS, DSC, organic PVs, and halide perovskites [13]. However, a similar comparison to the development of piezoelectric materials shows that the materials widely in use for commercial applications are still the ones developed in the 1970–1980s such as PZT, BaTiO_3_, etc. 

Piezoelectric materials convert mechanical energy to electrical energy by developing a dipole moment in them when mechanical stress is applied, and, vice versa, they become deformed when an electrical energy is applied. Our interest in piezoelectricity is based on two aspects: (a) it can be considered for both macro- and microscale energy harvesting since the harvestable sources are available in both ranges in the ambient environment and (b) the necessity of developing newer piezoelectric materials and fabrication methods to advance this technology further as the existing materials such as Barium Titanate and lead zirconate suffer from brittleness, lack of flexibility, and conformability, require high-temperature and high-pressure processing, and are usually bulky [14]. 

Recently, halide perovskites have been reported to possess promising piezoelectric properties in addition to their photovoltaic properties [15,16]. Hybrid halide perovskites are a family of materials that bind organic and inorganic components to a molecular composite. These materials are represented by a general chemical formula of ABX_3_, where A is an organic or inorganic cation, B is a divalent cation, such as Pb^2+^ or Sn^2+^, and X is a halogen, such as I^−^, Br^−^, or Cl^−^ (Figure 1a). Organic cations, A, can be methylammonium (MA) CH_3_NH_3_^+^, ethylammonium (EA) C_2_H_5_NH_3_^+^, formamidinium (FA) HC(NH_2_)_2_^+^, and even inorganic Cs [17,18]. The molecular structure of the hybrid perovskite metal halides consists of a three-dimensional (3D) metal halide framework, which encapsulates the polyatomic organic cations. The motion of the organic cations with large dipole moments in the metal halide cavity gives rise to unusual electrical properties in hybrid perovskites. This dipole moment (µ) is related to the symmetry of the organic cation, such as µMA^+^ >> µ FA^+^ > µ GA^+^ ~ 0, where MA^+^ = CH_3_NH_3_^+^ FA^+^ = HC(NH_2_)_2_^+^ GA^+^ = C(NH_2_)_3_^+^ [19]. 

Low-dimensional halide perovskites (general formula R_2_A_n−1_B_n_X_3n+1_) (Figure 1b) are also emerging as promising piezoelectric materials where the polarisation originates mainly from the contribution from the A cation as well as the symmetry breaking from the longer R cation [22,23]. Based on the symmetry, the hybrid perovskites can be classified as centrosymmetric, asymmetric, and non-centrosymmetric. Moreover, the degree of non-centrosymmetry can be tuned by applying an external stimulus of heat, electric field, photon energy, stress, etc. [10]. This tuning of non-centrosymmetry gives rise to many interesting properties of piezoelectric, pyroelectric, and high switchable dielectric constants in halide perovskites. Since halide perovskites are emerging as an important piezoelectric material to realise milli–microwatt power-generating devices, it is high time to consolidate the d_33_ coefficients reported for this family of materials in their bulk (crystals and powders) and thin-film formats and the methods routinely being used for these measurements. This review article is mainly focussed on the piezoelectric charge coefficient (d_33_) of various families of halide perovskites such as (a) 3D and (b) low-dimensional halide perovskites (2D and 1D). 

## 2. Piezoelectric Charge Coefficient (d_33_)

While developing new piezoelectric materials, one of the most important parameters to consider is its piezoelectric coefficient (d_ijk_) or the charge coefficient. Depending on the direct (mechanical to electrical) or converse piezoelectric effect (electrical energy to mechanical strain), the charge coefficients are defined as follows:$$d_{ij}=\frac{charge\;produced}{Applied\;stress}$$
$$d_{ij}^{,}=\frac{strain\;produced}{Applied\;voltage}$$

Associated with the $d_{ij}$ coefficient is the corresponding voltage coefficient g as well. Its definition is as follows:$$g=\frac{Electric\;field\;developed\;}{Applied\;Mechanical\;stress}=\frac{E}{T}$$

The relation between the charge coefficient ($d_{ij}$) and the voltage coefficient (g) is as follows: d=εg

When developing efficient mechanical energy harvesters, the efficiency of energy conversion is denoted by a parameter known as the coupling coefficient, k [24].
$$k^{2}=\frac{stored\;mechanical\;energy\;}{input\;electrical\;energy\;}=\frac{stored\;electrical\;energy\;}{input\;mechanical\;energy\;}$$

The value of k ranges from 0 to 1. k is related to the d_33_ coefficient as follows: $$k^{2}=dgY$$
where Y is the Young’s modulus. 

So, to have a higher coupling coefficient, the d_33_ coefficient needs to be higher. 

In addition to this, the stored electrical energy per unit volume is also directly proportional to the square of the d coefficient.
$$U_{ele.stored/volume}=\frac{1}{2}\frac{d^{2}T^{2}}{\varepsilon_{0}\varepsilon_{r}}$$
where T is the applied stress. In this review, we mainly focus on the d_33_ coefficients of the halide perovskites where the direction of the polarisation and the applied stress are along the same axis.

## 3. Measurement Methods of Piezoelectric Charge Coefficient (d_33_)

The existing methodologies for the d_33_ measurements are briefly summarised below: Quasi-static (Berlincourt method);Dynamic resonance;Laser interferometry;Piezoforce microscopy (PFM).

### 3.1. Quasi-Static Measurement Method for d_33_ Coefficient

The term quasi-static means the measurements are conducted slowly enough (almost statistically) at low frequencies so that the d_33_ values are measured under equilibrium conditions. The quasi-static measurement of the d_33_ coefficient is also widely known as the Berlincourt method after Don Berlincourt, who devised the first commercial d_33_ meter based on the principle of the quasi-static method. 

To understand the principle of the QS method, starting with the basic (direct) piezoelectric relationship is useful:(1)$$D=\varepsilon^{T}E+dT$$
where D is the electric displacement, d is the piezoelectric charge (strain) coefficient depending on the direct (converse) piezoelectric effect expressed in C/N or pm/V, $\varepsilon^{T}$ is the dielectric permittivity under constant stress conditions, and T is the stress expressed in N/m^2^. 

When no external electric field E is applied or the field is very small, Equation (1) can be written as
(2)D=dT

Here, D is expressed in C/m^2^ and T in N/m^2^. By rewriting the above equation, the following is found:(3)$$d=[\left(\frac{Q}{A}\right)÷\left(\frac{F}{A}\right)]=Q/F$$
where Q is the charge developed due to an applied force F and A is the active area. This equation shows that the d_33_ coefficient can be obtained by measuring the charge generated along the polar direction (z-direction in this case) in response to a force applied along the same direction. 

In the original ‘static’ method, as reported by Jaffe et al., two weights (to apply force) and a capacitor with a large capacitance can be used for the d_33_ measurement [25]. One weight (considered as the ‘dead weight’ and the bias force) and the shunt capacitor (connected in parallel or across with the piezo material for which the d_33_ measurement is needed) together will satisfy the boundary condition of zero electric field. Then, the second weight is applied/removed, and the transient voltage across the shunt capacitor due to the applied force bias (stress) can measure the charge developed on the DUT. From this, one can estimate the d_33_ coefficient of the device under test. The voltage developed during the introduction and removal of the second weight should match the same change in stress. Even though this static method is simple, the reliability is low due to the high possibility of external static charge involvement, charge generation due to the pyroelectric effect, and thermal drift, and in the case of ferroelectric-type samples, the neglection of D/E can further mislead the d_33_ value measured as the electric boundary conditions are violated [25,26,27].

The quasi-static method was developed to overcome the shortcomings of the application of a static load and eliminate the drift due to the pyroelectric charge. In the quasi-static measurements, a low-frequency oscillating and small-magnitude mechanical force is applied, and the corresponding charge output is measured. The charge developed is then divided by the magnitude of applied force to obtain d_33_. Here, the measurements are performed at a few hundred hertz, and this should be below the resonant frequency of the system so that the same formula as applied to the static cases is still applicable [27].

The small oscillating force is applied using a loudspeaker-type coil. The upper (1 KHz) and lower (10 Hz) limit of the frequency of the applied force is, respectively, determined by the mechanical resonance of the force-head unit and thermal drift. The DUT and the reference sample (usually) PZT are connected along the same line of action of force (Figure 2a) so that the same force is applied to the sample as well as the reference sample. To avoid the rattling of the sample and to keep it stable during the measurement, a static pre-load is usually applied. Knowing the capacitance and the d_33_ coefficient of the reference sample and the high sensitivity of the reference sample allows for the measurement of the force applied. The DUT and the reference sample electrical connections are typically as shown in Figure 2a. The constant field or short-circuit condition is achieved by connecting a large capacitor across the DUT (shunt capacitor) or by using a virtual earth amplifier. The d_33_ coefficient can then be determined by measuring the charge across the sample and reference or by taking the ratio of the respective voltages. In the latter case, taking the ratio of the voltage or (charge) across the sample to the reference gives the d_33_ coefficient. 

To ensure that the d_33_ coefficient is measured under equilibrium conditions or with the minimum error, attention needs to be paid to the input parameters such as the static pre-load, magnitude and frequency of the oscillating force applied, time-dependent effects, and proper calibration of the measurement system. The magnitude of the oscillating force is usually of the order of 0.1 N, and this relatively low force ensures that the DUT is in the linear regime (the charge produced is linear to the force). The frequency of the oscillating force field is usually 100 Hz, as above this and towards 1 kHz, there is interference from the resonance of the force head. Depending on the soft or hard piezoelectric materials, a response below 100 Hz can be frequency-dependent and is purely material-dependent. The low-frequency limit of 10 Hz is usually set by the thermal drift and the static charge dissipation to the surroundings. The purpose of the pre-load is to hold the sample in place and to establish the zero-field boundary conditions so that it does not move during the oscillating force. The range within which d_33_ values remain stable is typically below 10 N. 

### 3.2. Dynamic Resonance Method

The dynamic resonance method is also known as the frequency method, resonance–antiresonance, and double resonance method. This method can give a complete matrix of the material coefficients, such as the elastic compliance constants (S_ij_), piezoelectric voltage and charge constants (g_ij_ and d_ij_), electromechanical coupling (k_ij_) coefficient, capacitance, and dielectric constant [28,29]. This method works by the principle of natural and resonant frequency of samples of different sizes and shapes. The resonant frequency of a sample corresponds to the frequency of vibration of a system (electrical or mechanical in this context) at which it gives the maximum oscillatory response corresponding to an external excitation condition, and it happens when the external driving frequency matches with the natural frequency of the sample under test [27]. To conduct measurements using this method, an alternating electrical signal is applied at the ends of the piezoelectric sample, and the frequency of the excitation signal is swept by keeping its amplitude the same (reverse piezoelectric effect). At the resonance condition, the real part of the admittance becomes maximum, and the imaginary part is close to zero. This method is easier to understand considering the equivalent circuit of a piezoelectric device which consists of a parallel connection of static capacitance (one branch) and another branch representing the dynamic part (a series connection of L, C, and R), as shown in Figure 2b [27,28,30]. Depending on the sample geometry, a variety of vibration modes is possible by inducing resonance effects in different directions. There are recommended sample geometries for exciting the various modes [27]. An impedance analyser is used to carry out this measurement, and the admittance circle will give information about the series and parallel resonant frequencies f_s_ and f_p_, the resonant frequency f_r_, antiresonant frequency f_a_, maximum impedance frequency f_max_, and minimum impedance frequency f_min_, which in turn can be employed to extract the main tensor matrix component of the material coefficients of the piezoelectric crystal in vibration mode [28,29]. The disadvantage of this method is the requirement of samples of different shapes such as disc, plate, and cylinder with certain aspect ratios to extract the complete matrix coefficients. 

**Figure 2 materials-17-03083-f002:**
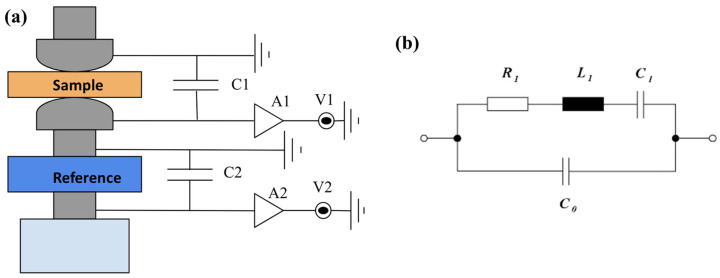
(**a**) Schematic of the Berlincourt method; (**b**) equivalent circuit of a piezoelectric transducer [31].

### 3.3. Laser Interferometry Method

Like the resonance–antiresonance method, the laser interferometry method also uses the reverse piezoelectric effect. In this method, after the application of a polarisation voltage, the nano–microrange of displacements generated through the piezoelectric effect is measured through high-resolution laser interferometry [27,29]. This method allows for the measurement of piezoelectric charge coefficients, without mechanical contact, and a thin-film expansion with a measurement resolution much better than 1 pm is possible. However, the method requires high accuracy in the construction and isolation of the measurement system from the parasitic vibration [32]. 

In a typical Michelson interferometer, for the measurement of the piezocoefficient, a reference and a probe beam constituted by a monochromatic wavelength λ are made to interfere, and the intensity of this interfering light beam is measured. The interference light intensity is given by the following:$$I={\left|E_{p\left(t\right)}+E_{r\left(t\right)}\right|}^{2}=I_{p}+I_{r}+2\sqrt{I_{p}I_{r}}\;cos\left(\frac{4\pi ∆d}{\lambda }\right)$$
where $I_{p}\;and\;I_{r}$ are the intensities of the probing and reference beams. ∆d is the optical path length difference between two beams [33]. The optical path length ∆d is related to the sample displacement. Corresponding to this displacement, an amplified output voltage signal is produced. The interferometry sensitivity parameter (IFS) is set to a specific value based on the displacement/unit volume (such as 10 nm/V or 50 nm/V). Using these parameters, the charge coefficient $d_{33}$ is calculated using the following relation [32]:$$d_{33}=\frac{\Delta l}{V_{in}}=\frac{IFS\times \Delta V}{V_{in}}$$
where Δl is the displacement, V_in_ the input voltage, IFS is the spectral sensitivity, and ΔV is the voltage. 

### 3.4. Piezoforce Microscopy (PFM)

PFM is used to study the piezo and ferroelectric properties of the materials at the nanoscale, and it measures the nanoscale topography simultaneously with the electromechanical coupling ($d_{33}^{\mathrm{eff}}$) in response to an electrical signal. 

It works based on the converse piezoelectric effect (Figure 3), and here, the voltage signal is applied to the sample and measures its mechanical extension at the nanoscale. 

The converse piezoelectric coefficient is $d_{33}^{\mathrm{eff}\;}=\frac{\;\mathrm{displacement}\;(pm)}{\;\mathrm{applied}\;\mathrm{voltage}\;(V)}=\frac{dS}{dV}$.

This technique is an alternate version of contact-mode Atomic Force Microscopy (AFM), and here, the AFM tip is in contact with the sample surface. PFM is used for topography imaging; it falls under the derivative imaging mode, and it is derivative of primary imaging. 

In PFM, the tip is conductive; here, an electrical signal is applied between the piezoelectric sample and tip. The applied electric field-induced mechanical strain response is associated with surface deformation, which reflects on surface topography. The lock-in amplifier compares the applied input signal $V_{in}=Acos(\omega t+\varphi )$ to the reference signal $V_{ref}=Bcos(\omega t)$, which are multiplied together to obtain a demodulation output.
$$V_{\mathrm{out}\;}=\frac{1}{2}ABcos(\varphi )+\frac{1}{2}ABcos(2\omega t+\varphi )$$

Here, A and B are input and reference signal amplitudes; ω, φ is the frequency and phase of both the input and reference signals. From this amplifier output, finally, the phase and amplitude images of the resultant piezoelectric sample are extracted.

The number of publications detailing the piezoelectric properties of halide perovskites is increasing rapidly, as shown in Figure 4. In the subsequent sections, the d*_ij_* coefficient of halide perovskites is summarised based on the sample format, such as thin films and crystals, and the measurement method, as depicted above. Since the Curie temperature (*T_c_*) is critical in identifying the various functional applications and enhanced power handling capacity of ferroelectric and piezoelectric materials and their integration with modern electronics, this information is included for the halide perovskites while listing the d_33_ values whenever this information is available.

## 4. d_33_ Coefficients of Halide Perovskites

### 4.1. d_33_ Coefficients of Halide Perovskite Crystals

Table 1 lists the d_33_ coefficients reported for the halide perovskite crystals. Recently, Zhao et al. reported a piezoelectric coefficient of 7 pC/N for MAPbI_3_ single crystals measured using a custom-made d_33_ meter. The ferroelectricity in MAPbI_3_ is attributed to the order–disorder transition of the MA^+^ cation, and it has a Curie temperature of 58 °C [35]. A slightly higher piezoelectric coefficient of 10.81 pV/m has been reported for MAPbI_3_ single crystals using PFM measurement. These MAPbI_3_ crystals also demonstrated pressure-induced enhancement in the photodetection properties, i.e., the piezo–phototronic effect [36]. Light-induced enhancement in the piezocoefficient from 2.0 to 4.31 pC/N as measured through the quasi-static method has been reported for MAPbBr_3_ single-crystal samples [37]. The light source used was a 405 nm laser with a power density of ~ 200 mW/cm^2^. A consistently higher d_33_ coefficient >10 pC/N measured by the quasi-static method has been observed for FA*_x_*MA_1–*x*_PbI_3_ (*x* = 0–0.1) single crystals [38]. A light-intensity-dependent d_33_ coefficient has been reported for thin MAPbBr_3_ single crystals. The d_33_ coefficient measured using custom-built d_33_ meter working on the principle of the quasi-static method increased from 20 pC/N to 87 pC/N as the light intensity increased from 4 mW/cm^2^ to 100 mW/cm^2^ [39]. A large d_33_ coefficient of 137 pC/N measured using the quasi-static method has recently been reported in a vacancy-ordered double halide perovskite (with the structural formula A_2_BX_6_) of TMCM_2_SnCl_6_ crystals where TMCM represents trimethylchloromethylammonium. In addition, 50% [SnCl_6_]^2–^ octahedrons are replaced by periodic vacancies in piezoelectric TMCM_2_SnCl_6_ in which the asymmetric TMCM^+^ takes the A sites [40]. The origin of piezoelectricity is attributed to the halogen-bond-mediated united movement of atomic displacement in [SnCl_6_]^2–^ octahedrons and the molecular rotation of the A cation TMCM^+^. 

A molecular ferroelectric and semiconducting 2D halide perovskite of (4-amino tetrahydropyran)_2_PbBr_4_ [(ATHP)_2_PbBr_4_] crystals has recently been reported as a promising piezoelectric material with a high piezoelectric d_33_ coefficient of 76 pC/N, as measured using the piezoforce microscopy (PFM). This material has a high Curie temperature of 503 K, higher than the conventional oxide perovskite of Barium Titanate [41]. Using the Berlincourt method, a large d_33_ coefficient of 1540 pC/N was obtained for the molecular ferroelectric solid solution with a morphotropic phase boundary (MPB) for the composition range (0 ≤ x ≤ 1) (TMFM)_x_(TMCM)_1–x_CdCl_3_ (TMFM, trimethylfluoromethyl ammonium; TMCM, trimethylchloromethyl ammonium) [42]. This high d_33_ coefficient is attributed to the existence of a morphotropic phase boundary (MPB) of monoclinic and hexagonal phases for this particular composition. Previously, the MPB between the ferroelectric tetragonal and rhombohedral phases was reported for conventional oxide perovskites such as PZT leading to its high d_33_ of ~590 pC/N for a 52/48 Zr/Ti ratio [26,43]. However, it is worth noting that the investigation of MPB in halide perovskites is in its very infancy. It is highly promising to note that a series of pb-free halide perovskite molecular ferroelectric crystals with TMCM, trimethylchloromethyl ammonium, as the A cation has shown d_33_ values ranging from 100 to 500 pC/N, comparable to that of conventional PZT, attracting research attention [44]. However, Chen et al. have noted that by introducing Br into TMCMC-CdCl_3_, it is possible to soften the lattice and simultaneously obtain a large voltage and charge coefficient. Thus, the authors reported a high d_33_ value of 440 pC/N measured using the quasi-static method and a g_33_ value of 6215 × 10^−3^ Vm/N [45]. Using the design principle of homochirality, Yang et al. designed piezoelectric and ferroelectric 2D perovskites of [R- and S-1-(4-Chlorophenyl)ethylammonium]_2_PbI_4_ with a d_33_ coefficient of 3 pm/V measured using PFM. These materials showed a bandgap of ~2.3 eV and good absorption in the visible spectral range [46]. Using DFT calculations, a one-dimensional enantiomorphic hybrid metal halide R/SMPCdCl_4_(R/SMP = R/S-2-methylpiperazine) with chiral properties was predicted to have high piezoelectric constants (16.71, 8.39, and 7.35 pC/N) [47]. By introducing chiral and rigid organic cations of α-phenylethylammonium, *R*-α-PEA^+^, and *S*-α-PEA^+^ into bismuth-based hybrid halides, chiral (*R*-α-PEA)_4_Bi_2_I_10_ and (*S*-α-PEA)_4_Bi_2_I_10_ crystals (centimetre-sized single crystals) were formed to yield an enhanced piezoelectric coefficient d_22_ of 32 pC/N, as measured using the quasi-static method [48].

Halogenobismuthates(III) with a piezoelectric effect are rarely reported. Recently, semiconducting and piezoelectric crystals of trimethylsufonium bismuth bromide [(CH_3_)_3_S]_3_[Bi_2_Br_9_] with a d_33_ coefficient of 18 pC/N measured using the quasi-static method were reported by Zhang et al. [49]. New hybrid lead-free metal halide (BTMA)_2_CoBr_4_ (BTMA = benzyltrimethylammonium) crystals with normal and shear piezoelectric responses (d_22_ and d_25_) of 5.14 and 12.4 pC/N, respectively, and with a non-perovskite crystal structure were synthesized by Guo et al. [50] In addition to the conventional octahedral perovskite structures (ABX_3_) mostly investigated, tetrahedral perovskites with the general formula (AM_2_X_5_) (A is a cation, M = Pb or Sn, and X = Cl, Br, or I) also seem to be promising for exploring the piezoelectric properties. Recently, Sahoo et al. reported a d_33_ coefficient of 72 pm/V for CsPb_2_Br_5_ microplates measured using the PFM method [51]. Single crystals of metal-free halide perovskite of N-methyl-N′-diazabicyclo [2.2.2] octonium–ammonium triiodide (MDABCO-NH4I_3_) have recently been reported to possess a d_33_ coefficient of 12.8 pm/V using the PFM method, and the promising potential of these materials for wearable applications were also demonstrated [52]. Halide double perovskites having a general formula of A_2_B^+^B^3+^X_6_ or A_2_B^4+^X_6_ (A = MA or Cs; B = Pb, Sn, Cu, or Bi; X = Cl, Br, or I) possess tuneable optoelectronic properties and better air stability compared to 3D halide perovskites. A comparison with three different Pb-halide-based and Sn-halide-based double perovskites showed negative d_33_ coefficients of MA_2_SnCl_6_ (d_33_ = −32.51 pC/N), MA_2_SnBr_6_ (−9.00 pC/N), and MA_2_SnI_6_ (−7.97 pC/N). These values were estimated using DFT calculations. The study also showed an increasing trend of d_33_ values with the size reduction in the halogen ions [53]. In contrast to this observation, an increasing piezoelectric response with an increase in halogen size was recently predicted using DFT calculations for CsGeX_3_ (X = Cl, Br, and I) as 0.731, 1.829, and 12.48 C m^−2^ for X = Cl, Br, and I, respectively [54]. 

**Table 1 materials-17-03083-t001:** List of d_33_ coefficients of halide perovskite crystals and the measurement method used. The Curie temperature (T_c_) is also included in the case of molecules for which this information was available.

Halide Perovskite Composition	d_ij_ Coefficient (d_33_ Unless Mentioned)	Measurement Method	Curie Temperature Tc(°C)	Reference
MAPbI_3_	7 pC/N	Quasi-static	58	Ref [35]
MAPbI_3_	10.81 pm/N	PFM		Ref [36]
FA_x_MA_1–x_PbI_3_ (x = 0–0.1)	10 pC/N	Quasi-static		Ref [38]
MAPbBr_3_	20 pC/N	Quasi-static		Ref [39]
TMCM_2_SnCl_6_	137 pC/N	Quasi-static	92	Ref [40]
(ATHP)_2_PbBr_4_	76 pC/N	PFM	230	Ref [41]
(TMFM)_x_(TMCM)_1–x_CdCl_3_ (0 ≤ x ≤ 1)	1540 pC/N	Quasi-static	100	Ref [42]
TMCMC-CdCl_3_,	440 pC/N	Quasi-static	100	Ref [45]
[R- and S-1-(4-Chlorophenyl)ethylammonium]_2_PbI_4_	3 pm/V	PFM		Ref [46]
(*R*-α-PEA)_4_Bi_2_I_10_ and (*S*-α-(PEA)_4_Bi_2_I_10_	32 pC/N (d_22_)	Quasi-static	197	Ref [48]
[(CH_3_)_3_S]_3_[Bi_2_Br_9_]	18 pC/N	Quasi-static	25	Ref [49]
(BTMA)_2_CoBr_4_	d_22_ 5.14 pC/N d_25_ 12.4 pC/N	Quasi-static		Ref [50]
CsPb_2_Br_5_	72 pm/V	PFM		Ref [51]
MDABCO-NH4I_3_	12.8 pm/V	PFM	90	Ref [52]

### 4.2. d_33_ Coefficients of Halide Perovskite Thin Films

Table 2 lists the d_33_ coefficients of halide perovskite thin films and the measurement method used. The first report on the d_33_ coefficients of halide perovskites was published in 2015 by Juan Bisquert et al. Using the PFM method, the d_33_ of MAPbI_3_ was measured to be 5 pm/V, which then increased to 25 pm/V upon white light illumination [55]. The incorporation of Fe^2+^ to partially replace Pb^2+^ has been found to profoundly enhance the d_33_ coefficient of MAPbI_3_ from 5 pm/V to 17.5 pm/V, as measured through PFM [56]. A comparison of the piezoelectric coefficient of MAPbI_3_ polycrystalline films on different substrates showed that by interfacing MAPbI_3_ thin films with PZT substrates the d_33_ coefficient measured using the PFM method enhanced from 0.3 pm/V to 4 pm/V by aligning the dipole moment of the MA cation [57]. Kim et al. reported the first all-inorganic and multilayered (with Cu metal) CsPbI_3_ thin films with a d_33_ coefficient of 15 pm/V measured using the PFM method [58]. After the poling process, the d_33_ value of the multilayered structure improved to 30 pm/V. Strain engineering has been successfully used by Kim et al. for CsPbBr_3_ films to modify the d_33_ coefficient from 9.3 pm/V for 0.75% tensile strain to 23.2 pm/V for a compressive strain of an equal amount [59]. A similar strain engineering mechanism has been applied to hybrid organic–inorganic thin films of MAPbI_3_, MAPbBr_3_, and MAPbCl_3_ to enhance the d_33_ effective coefficient estimated with the PFM technique, respectively, from 9.7 to 17.9 pm//V, 4.3 to 5.9 pm/V, and 2.7 to 3.4 pm/V. The origin of the piezoelectricity in cubic MAPbBr_3_ and MAPbCl_3_ has been attributed to the soft polarity modes and relatively low elastic modulus which modifies the organic–inorganic hydrogen bonding, lattice distortion, and ionic migration. Pb-free MASnI_3_ films with a d_33_ coefficient of 20 pm/V estimated using the PFM method were reported by Ippili et al., and this is higher than the d_33_~5pm/V commonly reported for the Pb-containing counterpart MAPbI_3_ [60]. By regulating the strain in the thin films, a series of CsPbX_3_ and their mixed halides were explored for their piezoelectric properties by Khan et al. They controlled the halogen atoms (I, Br) in the CsPbX_3_ (X = Br, Cl) lattice, and intriguing piezoelectric properties of CsPbBr=, CsPbBr_2_I, Cs2PbBr_2_I_2_, CsPbI_3_, CsPbCl_3_, CsPbCl_2_Br, and CsPbCl_2_I were reported. The reported d_33_ values range from 12 pC/N to 47 pC/N. The CsPbIBr_2_ showed the highest d_33_ coefficient of 47 pC/N, as measured using the vertical-PFM method [61]. 

### 4.3. Dimensional Tuning of Halide Perovskites and the d_33_ Coefficient

Recently, there has been increased interest in the dimensional tuning of halide perovskites to enhance the d_33_ coefficient [63]. A chiral, one-dimensional perovskite (*R*)-(−)-1-cyclohexylethylammonium)PbI_3_ has shown to be piezoelectric and ferroelectric, and a d_33_ value of 36 pm/V was measured, higher than that of the 3D hybrid halide perovskites. Another new one-dimensional BaNiO_3_-like organic−inorganic hybrid perovskite (thiazolidinium)CdBr_3_ in bulk pellet form has reported a d_33_ coefficient of 15 pc/N, as measured by the conventional quasi-static method [64]. Nanoparticles of FASnBr_3_ with an excellent d_33_ coefficient, as measured by PFM, have also recently been reported [65]. 

In contrast to the conventional oxide perovskites, the poling process is not necessary for halide perovskite-based ferroelectric/piezoelectric materials to induce polarisation, as these materials exhibit a self-poling effect [66]. This implies that the dipoles are orderly arranged during the crystallisation process, as has recently been reported by Tao et al. [48,51,67]. So, poling treatment is not routinely performed in these materials, which is an added advantage as it saves energy input, time, and cost. Based on the studies which have conducted the poling process, the applied poling field ranges from 3 kV/cm to 80 kV/cm [15,56,58,59]. These studies show that the poling process enhances the electrical output from the piezoelectric devices based on halide materials and enhances the d_33_ values. The origin of d_33_ enhancement is mainly ascribed to the increased BX_6_ octahedral distortion in the lattice and the off-centring of the B atoms [58,60,62]. Chen et al. [45] and Khan et al. [61] reported that applying a poling voltage of 3 V and 40 V during the PFM measurement was enough to observe the domain switching during the PFM measurement. 

The grain size effect on the piezoelectric properties of halide perovskites has not been exploited in detail so far. However, recently, Khan et al. [61] reported that a large grain size (~ 200 nm) is important for enhanced d_33_ values, and the authors attribute this to the easier displacement of domain walls in larger grains. The grain size information is not reported for all the halide perovskites listed in Table 1 and Table 2. However, based on the data available, the grain size of some of the piezoelectric halide perovskites (CsPbI_3_ [58], CsPbBr_3_ [59], MASnI_3_ [60], CsPbI_2_Br [61], and MDABCO-NH4I_3_ [52]) listed in Table 1 and Table 2 ranges from 200 to 500 nm.

Regarding the measurement methods of the d_33_ coefficient, a comparison of Table 1 and Table 2 shows that the crystal and bulk samples are measured using both the quasi-static and PFM methods. In the case of thin-film samples, mainly the PFM technique is used for d_33_ estimation; however, quasi-static measurements for thin-film piezoelectric materials are emerging, as recently shown by Garcia et al. [68]. Even though the Berlincourt meter is easy to use and fast in obtaining d_33_ values, there are some inherent challenges in using this method for thin-film piezoelectric coefficient measurements. Obtaining homogeneous uniaxial stress on thin films on a thick substrate and the possible interference of the transverse piezoelectric effect are considered the main issues [69]. Also, the static pre-load should be applied carefully since this damages the thin film’s electric contacts and the active material itself. Further, because of the simplicity of the set-up, there are several commercial systems without any general standard available, making the external comparisons largely variable and difficult, apart from the relative performance comparison from batch to batch [27]. Similarly, regarding the PFM measurement of thin films, this method mainly measures the d_33_ on a nano–micrometre scale and lacks the overall macroscale information of d_33_. In the PFM method, in some cases, even non-ferroelectric/non-piezoelectric materials have shown piezoresponse hysteresis loops and a 180° difference in phase due to several alternative mechanisms including charge injection and electrostatic effect, ionic motion, electronic transport, and humidity rather than the intrinsic ferroelectricity of the material. So, PFM results alone are not sufficient in concluding the ferroelectric/piezoelectric nature of the thin films. To overcome these challenges, some modifications of the PFM measurement technique such as the presence of an explicit threshold field as a piece of strong evidence for probing the intrinsic ferroelectricity in thin films are also proposed. Moreover, compared to the Berlincourt method, PFM uses an indirect piezoelectric effect, and the measurement requires complex and high-cost electronics, which should have vibration-free isolation to have stable and reliable d_33_ measurements [70]. 

Though the Curite temperature is not explicitly described here, the family of halide perovskite materials described in Table 1 and Table 2 shows that the T_c_ for these materials ranges from room temperature to 230 °C. This demonstration of piezoelectric properties at RT and above is particularly useful while designing high-power electronic devices and in the integration of these energy harvesters with disruptive technologies such as the Internet of Things (IoT). It is worth noting that the Curie temperature is relatively lower for halide perovskites compared to conventional oxide perovskites such as PZT (T_c_ = 350 °C) [71].

## 5. Other Piezoelectric Factors of Halide Perovskites

This review mainly focuses on the d_33_ coefficient of the halide perovskites. However, in the design of efficient mechanical energy harvesters, other piezoelectric factors also need to be considered. The two other most important factors are (i) the voltage coefficient (g_33_) and (ii) the electromechanical coupling coefficient k^2^, as defined in Section 2. Huang et al. previously reported a voltage coefficient (g_33_) of 980 × 10^−3^V.m/N for the vacancy-ordered double perovskite of TMCM_2_SnCl_6_ (where TMCM is trimethylchloromethyl ammonium) [40]. This voltage coefficient is much higher compared to conventional oxide-based piezoelectrics such as PZT (20 − 30 × 10^−3^ V·m/N) [72,73]. The two-dimensional halide perovskite of (4-amino tetrahydropyran)_2_PbBr_4_ [(ATHP)_2_PbBr_4_] crystals, mentioned in Section 4.1, has also been reported to have a high g_33_ of 660 × 10^−3^ Vm/N. Nanorods of mixed halide composition of the same 2D perovskite [(ATHP)2PbBr_2_Cl_2_] have shown an even higher g_33_ coefficient of 900 mVm/N and a d_33_ coefficient of 64.6 pC/N [74]. Using these reported d_33_ and g_33_ coefficients and the elastic Young’s modulus of 2D perovskites as ~15 GPa [75], we estimated the electromechanical coupling coefficient (k^2^ = dgY) as high as 0.8 for 2D halide perovskites. Following the same strategy, the estimation of k^2^ for 3D halide perovskites (e.g., MAPbBr_3_) gives a value less than 0.1 by taking the d_33_ as 20 pC/N (Table 1), the relative dielectric function as 22 [76], and the elastic modulus as ~20 GPa [75]. These calculations show the very promising potential of low-dimension halide perovskites for efficient mechanical energy harvesting. 

**Summary:** A comparison of the d_33_ values of the halide perovskite crystals and thin films, as listed in Table 1 and Table 2, clearly demonstrates the promising potential of halide perovskites for piezoelectric energy harvesting. The reported d_33_ values for some of the thin-film halide perovskites are higher than those of thin-film PZT [69] and the most commonly studied polymer piezoelectric PVDF [77]. The myriad of composition and dimension tuneability in halide perovskites along with simple solution processing and easy amenability to form thin films show the promising future of these molecules in piezoelectric energy harvesting and integration with the IoT as compact, portable, and distributed power sources.

## Figures and Tables

**Figure 1 materials-17-03083-f001:**
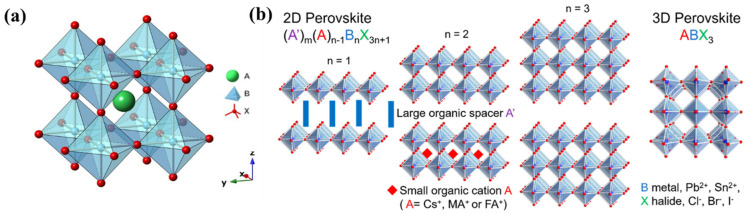
(**a**) Three-dimensional halide perovskite with the chemical formula ABX_3_ [20]. (**b**) Schematic illustration of the evolution from 2D perovskite to 3D perovskite with key components. Reproduced with permission from Ref [21].

**Figure 3 materials-17-03083-f003:**
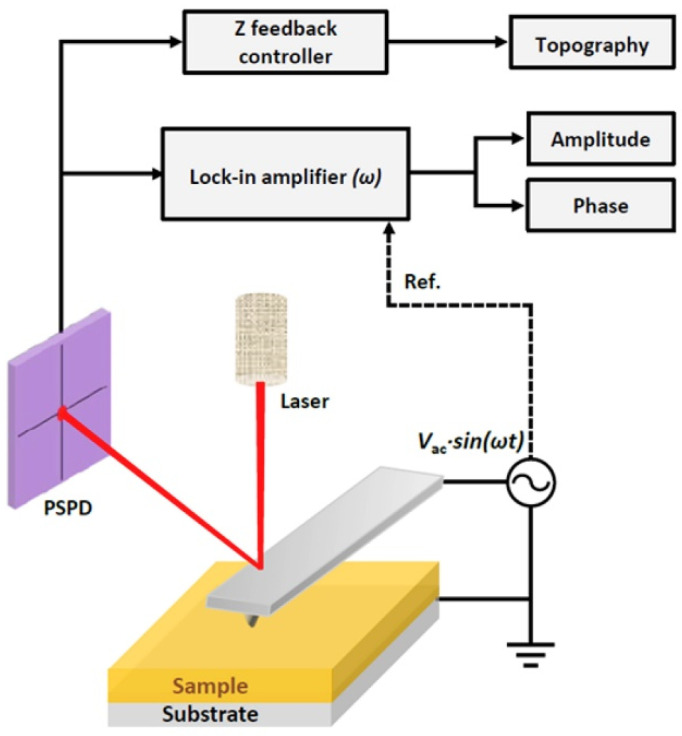
Working principle of piezoforce microscopy. Adapted from Ref [34].

**Figure 4 materials-17-03083-f004:**
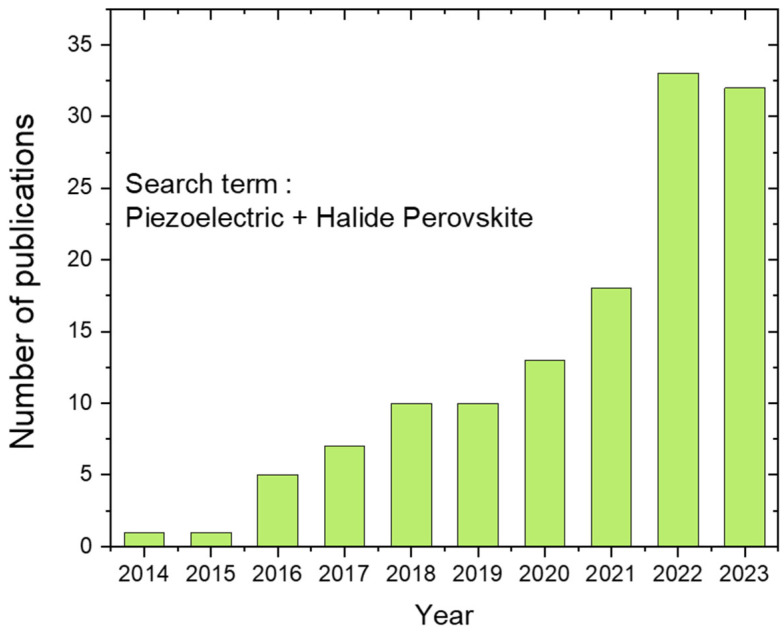
A graph showing the number of halide perovskite-based piezoelectric-related publications as a function of year.

**Table 2 materials-17-03083-t002:** List of d_33_ coefficients of halide perovskite thin films and the measurement method used. The Curie temperature (T_c_) is also included in the case of molecules for which this information was available.

Halide Perovskite Composition	d_ij_ Coefficient (d_33_ Unless Mentioned)	Measurement Method		Reference
MAPbI_3_	5 pm/N	PFM	58	Ref [55]
Fe: MAPbI_3_	17.5 pm/N	PFM	~44–47	Ref [56]
CsPbI_3_	15 pm/N	PFM		Ref [58]
CsPbBr_3_	23 pm/N	PFM	130	Ref [59]
MAPbCl_3_	3.4 pm/N	PFM		Ref [62]
MAPbBr_3_	4.9 pm/N	PFM		Ref [62]
MASnI_3_	20 pm/V	PFM	30	Ref [60]
CsPbI_2_Br	47 pm/V	PFM	130	Ref [61]

## Data Availability

No new data were created or analyzed in this study.

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
