# Peer review of "Piezoelectric Charge Coefficient of Halide Perovskites"

_materials, 2024, doi:10.3390/ma17133083_

Round 1
Reviewer 1 Report
Comments and Suggestions for Authors
The manuscript named "Piezoelectric charged coefficient of halide perovskites" written by Muddam et al. reports a detail analysis about the piezoelectric d33 coefficient on different type of Halide Perovskites measured by two different techniques. Although the manuscript is well written and easy to follow, there are some questions that authors must complete in order to be accepted.
For inorganic piezoelectric materials (e.g. BaTiO3, Pb(Zr,Ti)O3, (K,Na)NbO3) the maximization of the piezoelectric properties arises from a structural transition that occurs near a Polymorphic Phase Boundary (PPT) or a Morphotropic Phase Boundary (MPB). ¿Is this effect present in Halide Perovskites? Please explain.
Although the piezoelectric coefficient d33 is important, there are other factors (e.g. electromechanical coupling factor (kp), Piezoelectric voltage constant (g)) that must be taken into account for developing specific applications. Could the authors give a detailed reference about "other" piezoelectric factors, if available, in this kind of perovskites?
The temperature is another keypoint for the use of piezoelectric materials. Since most of the inorganic materials, except BaTiO3, have a Curie Point above 200°C, it facilitates the application of this materials in electronic devices. How is this effect observed in hybrid halide perovskite? Please explain.
What is the applied voltage range used in this kind of materials to induced a polarization?
Grain size is a significant characteristic for the piezoelectric activity. Could authors explain if there is any contribution of the grain size regarding the piezoelectric effect in halide hybrid perovskites. What is the grain size of the references showed in tables 1 and 2?
Comments on the Quality of English LanguageAdequate use of English
Author Response
Response letter attached

Reviewer 2 Report
Comments and Suggestions for Authors
The pros and cons of the different methods of measuring piezoelectric charge coefficients are manifold. To discuss them would not match the scope of this publication. But I would like you to raise awareness to the fact that Berlincourt meter has strong drawbacks regarding measurements on thin films, and that absolute d33 values from PFM have high uncertainties.
Also, I detected some mixing of the terms d33 and d33 eff, as well as different writings of d33 eff that require differentiation.

Author Response
File attached

Round 2
Reviewer 1 Report
Comments and Suggestions for Authors
The manuscript is accepted in its present form.